# Overview of Spontaneous Intracranial Hypotension and Differential Diagnosis with Chiari I Malformation

**DOI:** 10.3390/jcm12093287

**Published:** 2023-05-05

**Authors:** Wan Muhammad Nazief Bin Wan Hassan, Francesco Mistretta, Stefano Molinaro, Riccardo Russo, Giovanni Bosco, Andrea Gambino, Mauro Bergui

**Affiliations:** 1Radiology Department, Interventional Radiology Unit, Kuala Lumpur General Hospital, Kuala Lumpur 50586, Malaysia; 2Department of Neuroscience, Neuroradiological Unit, University of Turin, Azienda Ospedaliera Città della Salute e della Scienza Hospital, 10126 Turin, Italy; 3Department of Neuroscience, Stroke Unit, University of Turin, Azienda Ospedaliera Città della Salute e della Scienza Hospital, 10126 Turin, Italy; 4Department of Surgical Sciences, Radiology Unit, University of Turin, Azienda Ospedaliera Città della Salute e della Scienza Hospital, 10126 Turin, Italy

**Keywords:** spontaneous intracranial hypotension (SIH), Chiari malformation type I (CM1), myelographic imaging techniques

## Abstract

Spontaneous intracranial hypotension (SIH) occurs due to a leakage of the cerebrospinal fluid (CSF) lowering the pressure of subarachnoid space, mostly caused by a dural breach or discogenic microspur. As a result of less support provided by CSF pressure, intracranial structures are stretched downward, leading to a constellation of more or less typical MRI findings, including venous congestion, subdural effusions, brainstem sagging and low-lying cerebellar tonsils. Clinic examination and an MRI are usually enough to allow for the diagnosis; however, finding the location of the dural tear is challenging. SIH shares some MRI features with Chiari malformation type I (CM1), especially low-lying cerebellar tonsils. Since SIH is likely underdiagnosed, these findings could be interpreted as signs of CM1, leading to a misdiagnosis and an incorrect treatment pathway. Medical treatment, including steroids, bed rest, hydration caffeine, and a blind epidural blood patch, have been used in this condition with variable success rates. For some years, CSF venous fistulas have been described as the cause of SIH, and a specific diagnostic and therapeutic pathway have been proposed. The current literature on SIH with a focus on diagnosis, treatment, and differential diagnosis with CM1, is reviewed and discussed.

## 1. Introduction

The term spontaneous intracranial hypotension (SIH) describes a clinical condition characterized by an incapacitating orthostatic headache syndrome caused by a spontaneous leakage of the cerebrospinal fluid (CSF). More often than not, it is accompanied by stigmata of findings on intracranial magnetic resonance imaging (MRI). In 1939, Schaltenbrand wrote about the incidence of spontaneous CSF hypotension leading to various clinical symptoms, most notably positional headache. He also denoted that the entity itself had been discussed in the French literature twenty years earlier, using several terms such as ‘hypotension of the spinal fluid’ and ‘ventricular collapse’ [1]. The availability of MRI has raised the interest in SIH in the past few decades. While historically perceived to be a rare entity, SIH is now being diagnosed more often, with the incidence currently estimated at 5 per 100,000 annually [2]. However, considering the high occurrence of misdiagnosis—particularly at the initial stage of clinical presentation—the true incidence is likely higher [3]. Although rarely encountered in children, SIH may occur at any age, with the mean age of diagnosis being at around 40 years. Relating this condition, there is female predilection, with a female–male ratio of 2:1 [4,5,6]. Additionally, a higher incidence of SIH is observed in certain connective tissue disorders, particularly the Marfan syndrome and the Ehlers–Danlos syndrome [7,8]. According to the International Classification of Headache Disorders (ICHD) 3rd edition, SIH is diagnosed when there is a spontaneous onset headache, which develops in temporal relation to the occurrence of CSF hypotension (opening CSF pressure < 60 mm H_2_O); or evidence of CSF leakage (proven through imaging) [9]. However, more recent evidence has challenged the traditional conception of SIH as a condition caused by low CSF pressure.

Chiari malformation type I (CM1) is classified as a rare disease (ORPHA268882) and is historically defined as cerebellar tonsillar position greater than or equal to 5 mm below the level of the foramen magnum. Using this definition, imaging prevalence studies estimate the CM prevalence to be affect between 0.24 and 3.6% of the population. However, symptomatic cases of CM1 seem to have a very low incidence compared with CM1 imaging prevalence (radiological findings are frequently discovered accidentally due to the increased use of MRI in recent years). It is estimated that CM1 affects 1 in 1000 individuals symptomatically [10,11,12,13,14]. The prevalence of CM on MRI is higher in children and young adults than in the elderly. The tonsil position should not be expected to be static throughout life, and the different prevalence of CM between young and elderly individuals probably reflects changes that occur over the life course. The female sex is also associated with a higher prevalence of CM on imaging [12,13].

SIH shares some MRI features with Chiari malformation type I (CM1), especially low-lying cerebellar tonsils. Since SIH is likely underdiagnosed, these findings could be interpreted as signs of CM1, leading to a misdiagnosis and an incorrect treatment pathway.

## 2. Pathophysiology

The syndrome of SIH, in general, is a consequence of spontaneous CSF leaks along the neuraxis, almost invariably from the thecal sac of the spinal column [15,16,17]. Spontaneous leaks from the skull base are extremely rare [15,18]. Importantly, it has a distinct pathology from postdural puncture headaches or CSF hypovolemic conditions due to postoperative CSF loss or trauma [9].

Although the exact mechanism of spontaneous CSF leaks remain unknown, it is suspected that an underlying weakness of the spinal meninges is an imperative factor [19]. It creates a predisposition for the development of meningeal diverticula or perineural cysts, and subsequently spontaneous dural dehiscence. The presence of calcified degenerative thoracic disc protrusions may cause trivial traumatic events leading to progressive dural tears, which can be either ventral or lateral [7,19,20]. The CSF–venous fistula (CVF) is an important cause of SIH, which pose a diagnostic challenge [21]. Another rare, but interesting cause is the “nude nerve root” phenomenon, which describes the congenital absence of the entire nerve root sleeve [19]. The prevalence of SIH in certain connective-tissue disorders, particularly the Marfan syndrome, the Ehlers–Danlos syndrome (type II), and autosomal dominant polycystic kidney disease (ADPKD), are presumably due to the pre-existing dural fragility [7,8,18].

The old theories of SIH as a condition caused by CSF hypotension have been found to be inaccurate. A series of recent studies on larger samples by Luetmer et al. (2012), Yao et al. (2016), and Kranz et al. (2017), involving 388 patients, proved that only 21% to 55% of SIH patients had a low CSF opening pressure [22,23,24]. Following the data, it has been postulated that the reduction in CSF volume—rather than CSF pressure—is the primary pathogenetic factor [15,18,25]. Kranz et al. further advocated the concept of tissue compliance that determines the physiological relationship between pressure and volume [15]. Compliance with regards to the CSF component is highly dynamic and variable between individuals, as it is influenced by posture (upright vs. recumbent), body habitus, and the extent of the spinal epidural venous plexus [26,27,28]. It has also been demonstrated that the removal of CSF (via lumbar puncture) changes the spinal compliance due to the reduced spinal CSF volume [29]. Considering the fact that the rate of CSF loss is also different between individuals depending on mechanical factors, this may explain why not all SIH patients will have a low CSF opening pressure, despite a common unifying problem of low CSF volume.

It has been suggested that significant CSF loss disrupts the equilibrium between the volumes of intracranial fluid and the brain soft tissue, described as the Monro–Kellie hypothesis [30]. The Monro–Kellie hypothesis (with contribution by Cushing H. in 1926) indicated that with an intact skull, the sum of the volume of the brain, CSF volume, and intracranial blood volume is always constant. Hence, an increase in one leads to a compensatory reduction in the other (or both) of the remaining two variables [31]. Inversely, a decreased CSF volume requires the dilatation of the vascular spaces for volume compensation. Invariably, in this situation, the reactive hyperemia is reflected on the venous system due to its higher tissue compliance, as compared to the arterial side [30]. Venous hyperemia manifested in the venous sinuses, pituitary gland, and meninges (more conspicuously in pachymeninges) give rise to a constellation of cardinal findings in MR imaging.

In 2019, an international jury of experts awaiting new radiological criteria confirmed the historic radiological definition of CM1 (cerebellar tonsillar position greater than or equal to 5 mm below the level of the foramen magnum) (agreement 83.7%) [32].

Different mechanisms may underlie the pathogenesis of CM1, primarily the underdevelopment of the posterior fossa bony structures. Several studies have shown that many, but not all, patients with CM1 have a small posterior fossa [33].

Another mechanism is represented by the presence of hemodynamic alterations that increase intracranial pressure. According to some authors, the delayed opening of the membrane covering the outlet of the fourth ventricle during fetal development could cause transient obstructive hydrocephalus with a consequent hernia of the hindbrain and cerebellar tonsils [34].

CM1 should be differentiated from cerebellar tonsil herniation secondary to space-occupying lesions (hydrocephalus, arachnoid cysts, brain tumors), and termed ‘acquired tonsillar ectopia’ by an international jury of experts (agreement 95.8%) [32]. Acquired tonsillar ectopia has an identical shape to the shape of the tonsils associated with idiopathic CM1, and may also be secondary to the downward displacement of the central nervous system due to decreased intrathecal pressure (either secondary to the over-drainage of lumboperitoneal shunting or a spontaneous CSF leak) [32,34,35]. In the latter case, careful clinical assessment, ascertaining postural headache, and thorough radiological assessment for intracranial CSF hypotension can help differentiate acquired Chiari malformation from true Chiari I malformation, and guide appropriate treatment (see paragraph 6).

Although these mechanisms that act on the normal cerebellar tonsils, causing their deformation and herniation through the foramen magnum, are well understood, there is no consensus on the pathophysiology of CM1. For example, hydrocephalus has been proposed as both an etiologic cause and a consequence of a Chiari malformation.

## 3. Classification of SIH

Several classification systems have been proposed as guidance for clinical management, as well as for academic discussion [36,37]. The most widely used classification of SIH was introduced by Schievink et al. in 2016 [36]. This classification is based on the three recognized morphological types of leaks—namely the dural tear, the meningeal diverticulum, and the CSF–venous fistula (CVF)—in combination with the findings of a presence (or absence) of extradural CSF on spinal imaging [30,31] (Table 1).

## 4. Clinical Presentation of SIH

In a recent meta-analysis of 1694 SIH patients by D’Antona et al., headaches were almost invariably present—being reported in 97% of patients—and in the vast majority, they were orthostatic in nature [6]. This classic headache that characteristically worsens with an upright posture was most pronounced at the occipital region, although frontal and holocephalic headaches were also common. It may be attributed to the traction of the cranial nerves and the pain-sensitive dura mater, caused by the sagging of the brain in the CSF hypovolemia [3,38,39]. Typically, the headache is alleviated within 15–30 min after lying in a recumbent position [5,15]. However, the posture-related component of the headache may also vanish or dampen in a chronic condition [3,40].

Traction, distortion, or compression of some of the cranial nerves, certain structures of the brain, brainstem, mesencephalon, and diencephalon are believed to be the cause of many central nervous system manifestations, as well as the various cranial nerve palsies seen in this disorder [39]. Common associated symptoms include nausea/vomiting, neck pain/stiffness, tinnitus, dizziness, and hearing disturbances. Apart from that, there are a myriad of less frequently reported presentations which include other visual symptoms (photophobia, diplopia, blurred vision, nystagmus, visual loss), back pain, cognitive symptoms (including cognitive impairment, behavioral changes, memory, slow thinking), ear-related symptoms (vertigo, aural fullness, hyperacusis, or unspecified), reduced level of consciousness, and movement disorders (gait disorder, ataxia, dysarthria, tremor, bradykinesia, poor balance) [6].

Other rare, but peculiar symptoms, such as galactorrhea and hyperprolactinemia, have been attributed to the stretching or distortion of the pituitary stalk and/or hypothalamic region [41,42]; meanwhile, the dilatation of the epidural venous plexus or traction and compression of nerve roots are speculated to be the cause of radicular symptoms and incontinence [43].

Other less common, innumerable symptoms have also been described, including dysgeusia, sleepiness, other cranial nerve palsy, fever, fatigue, vocal tics, convulsions, facial spasms/numbness/pain, and dysphagia [6].

## 5. SIH: Diagnostic Workup and Imaging Strategy

As lumbar puncture is an invasive procedure, in addition to the fact that only about half of the demographic shows a low CSF opening pressure, imaging emerges as the key aspect in the management of SIH. Furthermore, CSF analysis often exhibits considerable variability and is nonspecific for SIH [18].

Imaging plays a pivotal role both in confirming the diagnosis of SIH and localization of the leak point, which allows for the subsequent targeted therapy. A head MRI is the mainstay of initial imaging of SIH, as it is the most sensitive tool in detecting the signs of a CSF leak [6]. Nevertheless, while useful for confirming the diagnosis of SIH, brain MRIs are usually unable to localize the exact site and morphology of the CSF leak. Often, they need to be followed by other myelographic imaging techniques which have the advantage of depicting the direct evidence of CSF leakage (Figure 1), rather than the indirect signs seen with a cranial MRI [20,28].

Several myelographic imaging techniques have been developed in response to the daunting task of identifying the site of the CSF leakage, which can be anywhere along the neuraxis and with various rates of leakage. These include a spinal MRI, computed tomography myelograms (CTM), dynamic CTM, digital subtraction myelography (DSM), MR myelography, and radionuclide cisternography. Choosing the correct myelographic imaging technique from an array of imaging modalities available today is fundamental for the successful treatment of SIH.

A proposed imaging strategy starts with an MRI of the brain, with a contrast to look for signs of a CSF leakage [20,38,44,45]. Patients showing a high possibility of SIH features should be subjected to a non-directed epidural blood patch (EBP). Those whose head MRI findings were equivocal would benefit from a spinal MRI, which is a non-invasive investigation. A spinal MRI is also indicated in patients who are refractory to non-directed EBP. The findings of a spinal longitudinal extradural collection (SLEC) within the spinal MRI would determine the best myelographic techniques for further delineation of the leaking site. Either DSM or CTM/dynamic CTM performed in certain recommended positions demonstrate a higher yield of CSF leak point detection [45]. If no leak is identified, further exploration with nuclear medicine myelography or MRI myelography (with intrathecal Gadolinium) may reveal an intermittent leak or very slow leaks.

### 5.1. Head MRI

Numerous cranial MRI signs have been described and are linked with abnormalities caused by a CSF leak. Due to the CSF volume depletion, features of “brain sagging” may be evident in an MRI, characterized by the descent of cerebellar tonsils (mimicking Chiari type I malformation) (Figure 2), as well as the descent/distortion of the brainstem, mesencephalon, and diencephalon, with a flattening of the anterior pons. Additional findings include the obliteration of the basal cisterns (e.g., prepontine, perichiasmatic), flattening of the optic chiasm, and crowding of the posterior fossa [18,44]. Several specific values and definition have been used for a more qualitative evaluation of these sinking brain features in an MRI, including the pontomesencephalic angle, suprasellar cistern distance, prepontine cistern distance, venous–hinge angle, mamillopontine distance, pituitary height, tonsillar herniation length (relating to McRae line), and the area cavum veli interpositi [45].

Other important signs on a cranial MRI are derived from the reactive intracranial venous hyperaemia as a response to CSF hypovolemia. Venous engorgement is reflected on the distended dural venous sinuses, as well as the inferior intercavernous sinus. It is also responsible for the enlargement of the pituitary gland. The presence of subdural fluid is also part of the intracranial volume compensation in CSF loss [18]. However, the most sensitive intracranial sign pointing toward SIH is a smooth and diffuse pachymeningeal enhancement in a post-gadolinium MRI [6,20,45] (Figure 3A,B). This is due to the absence of the blood brain barrier (BBB) in pachymeningeas, opposed to the leptomeningeal layer, which permits the contrasted media leakage [46,47]. Apart from these, superficial siderosis has also been reported [45].

Nevertheless, no specific sign can be considered pathognomonic of the disease, but rather a constellation of typical MRI findings that allow for the confident diagnosis of SIH. In response to the challenge, Dobrocky et al. proposed a scoring system—termed the Bern SIH score—integrating six most relevant brain MRI findings which deliver the highest diagnostic accuracy [45]. Pachymeningeal contrast enhancement, engorgement of the venous sinuses, and an effacement of the suprasellar cistern (measuring ≤ 4.0 mm) were shown to be the most important discriminating features between SIH patients and normal controls. These features counted as the major criteria and were allocated with 2 points each. Subdural fluid collections, prepontine cistern effacement (measuring ≤ 5 mm), and a mamillopontine distance of ≤6.5 mm were the minor criteria and are weighted 1 point each. Patients with overall scores of 5 or more are classified as having a high possibility of a spinal CSF leak. Total scores of 3 to 4 are rated intermediate, while scores of 2 points or less are regarded as having a low probability for a spinal CSF leak [45]. The Bern SIH score can be used to stratify the probability of SIH and triage patients who should be promptly considered for targeted therapy, or those who may benefit from more invasive myelographic examinations [45].

### 5.2. Spine MRI

A spinal MRI is often complementary to a cranial MRI at the initial investigation of SIH. While it is rarely able to depict the exact site of leakage, it is useful for identifying the presence of extradural CSF pooling—termed spinal longitudinal extradural collection (SLEC)—which is a specific sign of SIH with dural mechanical tears along the thecal sac [6,20,44] (Figure 3C). SLEC is seen in 60–90% of SIH patients on a T2-weighted MR sequence [28,38,48,49]. The collections are usually diffuse and spread over multiple vertebral segments, away from the site of the leak. Hence, the exact site of leakage is often not identifiable by virtue of the epidural collection alone [20,28].

Nonetheless, the presence of SLEC (i.e., SLEC positive; or SLEC-P) can predict the underlying pathomechanism of the spinal CSF leak [44]. Type 1 CSF leaks, particularly type 1a, always demonstrate SLEC-P. Type 2 CSF leaks are also most often SLEC-P, except when the dural tear of the nerve root sleeve is very distal. Type 3 CSF leaks (CSF-venous fistula) demonstrate a negative SLEC (SLEC-N), as the CSF drains directly into abnormal venous channels without accumulating within the extradural space [38,44]. Further imaging techniques to pinpoint the site of the leakage can be tailored according to SLEC findings.

### 5.3. Digital Subtraction Myelography (DSM)

In most cases, it is not necessary to identify the site of the CSF leakage, because SIH often resolves with conservative management or after a non-directed epidural blood patch (EBP) [28,47]. However, a substantial number of patients experience recurrent symptoms, or a refraction to treatment with EBP. In this group of patients, localizing the leak point would allow for a more precise intervention, such as a directed EBP, glue embolization of CSF–venous fistulas, or a microsurgical repair of the leak (Figure 1).

The aforementioned dichotomy of SIH patients into the SLEC-P and SLEC-N groups based on spinal MRI serves as a reliable guide to choose the most effective patient position during a DSM [44]. Patients with SLEC-P would undergo a DSM or computed tomography myelograms (CTM) in prone position, with their head tilted down to enhance the prospect of identifying a ventral leak [20,38]. Particular attention should be given to the cervical and thoracic regions due to the higher incidence of fast CSF leaks at these sections [20,50]. SLEC-N patients would benefit from the procedure in a recumbent position, as theoretically, the leaks are located more laterally [38,44]. This position will promote the visualization of the contrast leak, as the dural defect is on the dependent aspect of the thecal sac. A recent study by Kim et al. showed 53% sensitivity in identifying the CSF leak point in SLEC-N SIH patients using recumbent position DSM [51].

Although DSM offers excellent spatial and temporal resolutions, it requires total patient cooperation to obtain the desired result. This is especially crucial at the cervical and thoracic regions due to the overlying soft tissue, shoulder bones, lungs, and possible motion due to an uncomfortable position. Owing to the possibility of having a bilateral leak, a DSM of the right and left recumbent position necessitates two examinations in separate sessions, which can be troublesome for patients [51].

### 5.4. CT Myelography

Other than a DSM, CT myelography (CTM) is the test of choice for the investigation of a suspected SIH at many centers [20]. This is due to its logistic advantages, such as a wide availability and familiarity among radiologists, as well as the ability to provide good spatial resolution with infrequent technical artifacts. An intrathecal administration of iodinated contrast media allows for the visualization of CSF, including the leaked CSF within the epidural space and intravenous leak in the CSF–venous fistula [52] (Figure 1D). The presence of causative degenerative spinal changes are readily imaged in excellent spatial resolution. The CSF opening pressure can also be measured during the procedure.

However, CTM has significant limitation in terms of temporal resolution, which can affect its diagnostic ability. The optimal imaging time from myelographic contrast media injection is speculative due to the variable rate of the CSF leak. In high-flow CSF leaks, the contrast media might have spread extensively over multiple spinal levels, rendering localizing the leaking point impossible. On the other hand, if the rate of the CSF leakage is sufficiently slow, the leak may not be visualized [20]. Some technical modifications have been implemented to improve the temporal resolution of CTM, including the use of CT fluoroscopy combination and dynamic CTM [22,53,54].

### 5.5. Radioisotope Cisternography

Radionuclide cisternography using a intrathecal injection of the Indium-111 radioisotope provides the solution for the recognition of very slow or intermittent leaks not detected in other imaging modalities. However, it provides a relatively poor spatial resolution, with moderate sensitivity and specificity [20].

## 6. Differential Diagnosis between SIH and Chiari Malformation Type I

Low-lying cerebellar tonsils can lead to misdiagnosis of a Chiari malformation type I (CM1) (Figure 2). Although both Chiari malformation type I (CM1) and SIH can cause chronic headaches, CM1 generally does not cause an abrupt orthostatic chronic daily headache, but rather causes headaches in the neck/occipital area, often as nonprogressive short attacks of ‘cough headaches’ [55]. As described above, headaches with orthostatic features are the hallmark of SIH, and are most recognizable within thirty minutes of standing and should be alleviated within 15–30 min after lying in a recumbent position, especially early in the disease course [6,7,8,9,15,40]. SIH can also cause a ‘cough headache’, but it is usually progressive in nature and can be associated with an underlying daily headache [56]. However, variability in clinical presentation for both SIH and CM1 may confound the diagnosis.

Distinguishing SIH from CM1 is crucial because the treatment for these two conditions is very different, as CM1 treatment involves suboccipital craniectomy, while treatment for SIH involves invasive interventional procedures or spinal surgery. A misdiagnosis can lead to delays in care and incorrect treatments, such as unnecessary surgery or procedures, with a needless risk of surgical or procedural complications.

According to a study by Houk et al., measures of midbrain sagging, including cut-off values for slope of the third ventricular floor and pontomesencephalic angle, may help discriminate CM1 from SIH [57]. Moreover, low-lying cerebellar tonsils from SIH should maintain the normal tonsillar rounded shape [58]. When present, pachymeningeal enhancement and other MRI features, including pituitary hyperemia, subdural collections, and venous engorgement, may be a useful means of distinguishing SIH from CM1. However, these features are not always present (pachymeningeal enhancement is absent in 15% of SIH cases) [59]. On the other hand, some features associated with CM1, including syringomyelia and tethered cord, are not commonly observed in SIH [60]. In addition, evidence has suggested that a reduced clivus volume and an increased sphenoid sinus volume are associated with CM1 [61].

Therefore, in CM1 differential diagnosis, false tonsil descent due to intracranial hypotension must be excluded via a clinical pattern, MRI pattern, and contrast-enhanced MRI [32]. The initial diagnostic work-up should include a head MRI (with and without contrast) and a full spine MRI to evaluate all possible imaging features of the two conditions (Figure 2 and Figure 3).

## 7. Treatment and Prognosis of SIH

Large-scale data and randomized trials are still lacking to guide the standard management of SIH. Current practice of conservative treatment, such as bed rest, high oral fluid intake, caffeine, and an abdominal binder, are often the first-line therapy [47]. Subsequently, non-directed EBP is the treatment of choice if all conservative measures fail [18]. In non-directed EBP, patients’ autologous blood is injected into the epidural space, usually at the lumbar region. This will induce fibrin clot formation at the site of the dural tear, acting as a “patch” to stop the leakage. Symptomatic relief is sometimes immediate, with the majority of patients only requiring one or two sessions of EBP to achieve symptomatic relief [6,47].

For patients with recurrent symptoms after non-directed EBP, further investigations to identify the exact site and type of the CSF leak are necessary. A more targeted approach to the causative pathology can be offered, such as a directed EBP, microsurgical repair of the leak or endovascular treatment for CVF [62,63,64]. The latter is a relatively new treatment of CVF and can be performed by venous embolization of the suspected fistula site: a diagnostic and therapeutic work-up have recently been proposed, showing their feasibility and short-term effectiveness [65].

One complication of a successful CSF leak treatment is the development of rebound high intracranial pressure, which results in reverse orthostatic headaches; the high pressure is generally self-limiting.

## 8. Conclusions

CSF leaks in SIH (due to a dural breach or CSF–venous fistula) lower the pressure of the subarachnoid space, leading the intracranial structures to be stretched downward. In these cases, since SIH is likely underdiagnosed, low-lying cerebellar tonsils could be interpreted as signs of CM1, leading to a misdiagnosis and an incorrect treatment pathway. Treatment for these two conditions is very different, as CM1 treatment involves suboccipital craniectomy, while treatment for SIH involves invasive interventional procedures or spinal surgery. A misdiagnosis can lead to delays in care and incorrect treatments, such as unnecessary surgery. The initial diagnostic work-up for the differential diagnosis is crucial and should include a head MRI (with and without contrast) and a full spine MRI to evaluate all possible imaging features of the two conditions.

## Figures and Tables

**Figure 1 jcm-12-03287-f001:**
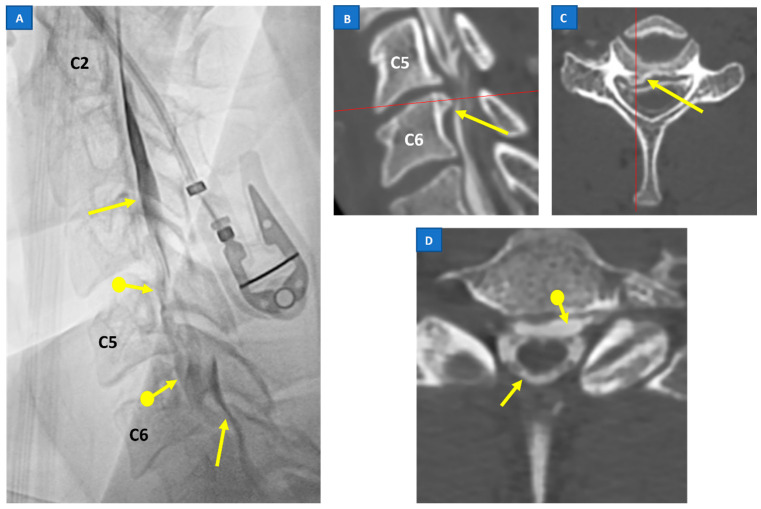
Prone myelography (**A**) showing a regular ventral profile of the dural sac (arrows) with anterior extravasation at C5–C6 into the epidural space (arrows with dot). Myelo-CT ((**B**,**C**): sagittal and axial planes) shows the presence of a C5–C6 disc herniation, with an osteophyte (bone spur) of the upper somatic margin of C6 (arrows), in the right paramedian site, probable cause of the dural tear (the red line indicates in image (**B**) the axial plane of image (**C**) while in image (**C**) the sagittal plane of image (**B**)). Myelo-CT (**D**) also shows the presence of contrast in the epidural space (arrow with dot), anterior to the dural sac (arrow).

**Figure 2 jcm-12-03287-f002:**
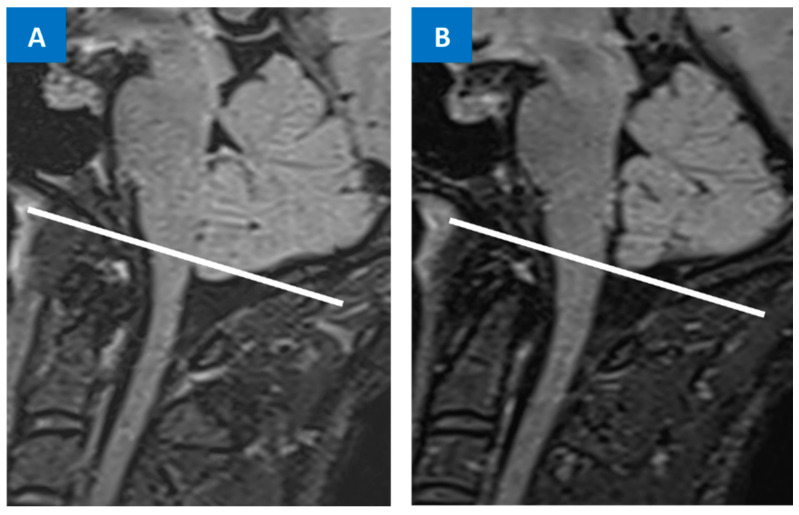
FLAIR sequence MRI showing low-lying cerebellar tonsils, which can lead to a misdiagnosis between SIH and Chiari malformation type I (**A**). Other MRI findings from the same patient are visible in Figure 3: clinical–radiological diagnosis of SIH is made and the patient is treated with an EBP. Resolution of the low-lying cerebellar tonsils was observed at the 12-month MRI follow-up (**B**).

**Figure 3 jcm-12-03287-f003:**
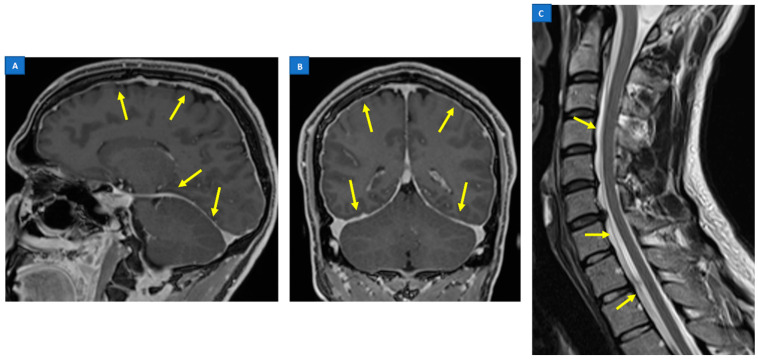
Smooth and diffuse pachymeningeal enhancement (arrows) in a post-gadolinium MRI ((**A**,**B**): sagittal and coronal planes): the most sensitive intracranial sign pointing toward SIH. Cervicodorsal spine MRI (**C**) showing the presence of a spinal longitudinal extradural collection (SLEC) (arrows), which is a specific sign of SIH with dural mechanical tears along the thecal sac.

**Table 1 jcm-12-03287-t001:** Classification of SIH by Schievink et al. [33].

	Morphological Type of Leak	Location of Leak
Type 1 (60% of cases) - 1a - 1b	Dural tear Dural tear	Ventral dura Posterolateral dura
Type 2 (20% of cases)		
- 2a	Simple single or multiple meningeal diverticula	Lateral dura
- 2b	Complex meningeal diverticula or dural ectasia	Lateral dura
Type 3 (20% of cases)	Direct CSF-venous fistula	Distal nerve root sleeve
Type 4	Indeterminate origin	

## Data Availability

Not applicable.

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
