# Peer review of "Overview of Spontaneous Intracranial Hypotension and Differential Diagnosis with Chiari I Malformation"

_jcm, 2023, doi:10.3390/jcm12093287_

Round 1

Reviewer 1 Report

Intracranial hypotension is the main question addressed by the research and consider the topic original and  relevant in the field.

The manuscript is clear and relevant to the field and is presented in a well-structured manner. The manuscript has a scientific basis.

Fiigures, tables, images and schemes are appropriate, they properly show the data and they easy to interpret and understand. 

The data were interpreted appropriately and consistently throughout the manuscript if the topic of this article about intracranial hypotension. However, it is necessary to write more about Chiari I malformation:

·    The topic of this article is about spontaneous intracranial hypotension and Chiari I malformation. Please add information on the prevalence of intracranial hypotension  and hypertension in Chiari I malformation patients.

·    In pathophysiology, you explain the causes of Intracranial hypotension in many diseases, except for Chiari I malformarion. Please add information. In additional, in most cases of Chiari anomaly we find intracranial hypertension, but there are many reports of intracranial hypotension. Please could you add information and pathogenic explanation for this.

·    In the classification. since we are talking about Chiari I malformation, can you add information about he most common types of CSF in Chiari malformation I. It will be more informative for the reader if the classification is described by the scheme.

There are  no discussion and conclusion. 

In literature there is a similar review published recently but this current review still relevant and of interest to the scientific community. (Kumar R, Cutsforth-Gregory JK, Brinjikji W. Cerebrospinal Fluid Leaks, Spontaneous Intracranial Hypotension, and Chiari I Malformation. Neurosurg Clin N Am. 2023 Jan;34(1):185-192. doi: 10.1016/j.nec.2022.08.012. Epub 2022 Nov 3. PMID: 36424060)

50% of submitted references are more than 5 years old.

It will be more informative for the reader if the classification is described by the scheme.

Figures under the numbers 1, 2, 3 are not presented according to the rules. The letters (a) (b) (c) must be in bold and below the figures. Better look at the MDPI. log rules.

Correct position of reference no. 11 and no. 31.

Author Response

Dear Editor, Dear Reviewers,
We would like to thank you for your constructive comments and suggestions regarding our manuscript we recently submitted to “JCM”. Below you will find the answers to your comments.

Q: “The topic of this article is about spontaneous intracranial hypotension and Chiari I malformation. Please add information on the prevalence of intracranial hypotension and hypertension in Chiari I malformation patients.
In pathophysiology, you explain the causes of Intracranial hypotension in many diseases, except for Chiari I malformarion. Please add information. In additional, in most cases of Chiari anomaly we find intracranial hypertension, but there are many reports of intracranial hypotension. Please could you add information and pathogenic explanation for this”.
A: We are very grateful for your observation! In the revised version of the manuscript, we described the pathophysiology of Chiari I malformation, focusing on intracranial hypotension and/or hypertension. We found no studies reporting the specific prevalence of intracranial hypotension and/or hypertension in CM1 patients.

Q: “In the classification, since we are talking about Chiari I malformation, can you add information about the most common types of CSF in Chiari malformation I. It will be more informative for the reader if the classification is described by the scheme”.
A: Thank you for the suggestion. All causes of SIH can lead to a downward displacement of the central nervous system due to decreased intrathecal pressure underling an "acquired" Chiari malformation. The case rate for the type of SIH is reported on the manuscript. To our knowledge there are no specific data in the literature on the most common types of SIH in Chiari I malformation.
The classification of SIH is described by a scheme in the revised version of the manuscript.

Q: “Figures under the numbers 1, 2, 3 are not presented according to the rules. The letters (a) (b) (c) must be in bold and below the figures. Better look at the MDPI. log rules.”.
A: Thanks for the correction, letters are in bold and captions are added below the figures in the revised manuscript.

Q: “Correct position of reference no. 11 and no. 31”.
A: Thanks for the correction, we've fixed it.

Q: “However, there is merely one paragraph on the differential diagnosis between CM1 and SIH. The whole text refers to the SIH more that to the CM1. Therefore, I suggest rewriting the paper into the review of the SIH. In its current form, the title is misleading. Moreover, the paper lacks conclusions. Adding that paragraph would be appreciated.”.
A: Thank you for the pertinent consideration. We changed the title to “Overview on spontaneous intracranial hypotension and differential diagnosis with Chiari I malformation” to clarify that the main topic of the manuscript is SIH with an important focus on the differential diagnosis with CM1. In our view the differential diagnosis between these two conditions is an important topic and the aim of this study is to make it clearer.
Moreover, we added a paragraph with the conclusions.

Reviewer 2 Report

Hassan et al. performed a literature review that focused on the comparison between spontaneous intracranial hypotension (SIH) and Chiari malformation type 1 (CM1). The paper remains well-written, remains clear, and concise. References are correctly chosen. Figures are properly selected and described. However, there is merely one paragraph on the differential diagnosis between CM1 and SIH. The whole text refers to the SIH more that to the CM1. Therefore, I suggest rewriting the paper into the review of the SIH. In its current form, the title is misleading. Moreover, the paper lacks conclusions. Adding that paragraph would be appreciated. 

Author Response

Dear Editor, Dear Reviewers,
We would like to thank you for your constructive comments and suggestions regarding our manuscript we recently submitted to “JCM”. Below you will find the answers to your comments.

Q: “There is merely one paragraph on the differential diagnosis between CM1 and SIH. The whole text refers to the SIH more that to the CM1. Therefore, I suggest rewriting the paper into the review of the SIH. In its current form, the title is misleading. Moreover, the paper lacks conclusions. Adding that paragraph would be appreciated.”.
A: Thank you for the pertinent consideration. In the revised version of the manuscript, we have better described the Chiari I malformation and in particular its pathophysiology, focusing on intracranial hypotension and/or hypertension. Moreover, we changed the title to “Overview on spontaneous intracranial hypotension and differential diagnosis with Chiari I malformation” to clarify that the main topic of the manuscript is SIH with an important focus on the differential diagnosis with CM1. In our view the differential diagnosis between these two conditions is an important topic and the aim of this study is to make it clearer.
Moreover, we added a paragraph with the conclusions.